# Infection and Burn Injury

**Edward J. Kelly** [1,2,*], **Mary A. Oliver** [1,2], **Bonnie C. Carney** [1,2,3] and **Jeffrey W. Shupp** [1,2,3]

1     Firefighters' Burn and Surgical Research Laboratory, MedStar Health Research Institute,
Washington, DC 20010, USA; mary.a.oliver@medstar.net (M.A.O.); bonnie.c.carney@medstar.net (B.C.C.);
jeffrey.w.shupp@medstar.net (J.W.S.)

2     The Burn Center, Department of Surgery, MedStar Washington Hospital Center, Washington, DC 20010, USA

3     Department of Surgery and Biochemistry, Georgetown University School of Medicine,
Washington, DC 20057, USA

\*     Correspondence: Edward.J.Kelly@Medstar.net

**Abstract:** Burn injury is debilitating and among one of the most frequently occurring traumas. Critical care improvements have allowed for increasingly positive outcomes. However, infection, whether it be localized to the site of the wound or systemic in nature, remains a serious cause of morbidity and mortality. Immune suppression predisposes the burn population to the development of invasive infections; and this along with the possibility of inhalation injury puts them at a significant risk for mortality. Emerging multi-drug-resistant pathogens, including *Staphylococcus aureus*, *Enterococcus*, *Pseudomonas*, *Acinetobacter*, *Enterobacter*, and yeast spp., continue to complicate clinical care measures, requiring innovative therapies and antimicrobial treatment. Close monitoring of antimicrobial regimens, strict decontamination procedures, early burn eschar removal, adequate wound closure, proper nutritional maintenance, and management of shock and resuscitation all play a significant role in mitigating infection. Novel antimicrobial therapies such as ultraviolet light, cold plasma and topical antiseptics must continue to evolve in order to lower the burden of infection in burn.

**Keywords:** burn; infection; MRSA; trauma; MDRO; HAI; sepsis; CAUTIs; CLABSIs

## 1. Introduction

Despite significant improvement in the morbidity and mortality associated with burn injuries, infection remains one of the most common and serious complications in the care of the burn patient [1,2]. In order to minimize infection, clinical interventions have focused on early burn excision and grafting, novel wound dressings, adequate ventilation and hemodynamic support, improved nutrition, and suitable antibiotic administration [3–6]. However, 42% to 65% of all deaths in patients with burns are attributable to infection [7,8]. Specifically, burn wound infections complicate 1.8% of all burn admissions and urinary tract infections, cellulitis and pneumonia are the most common fatal infections [2,7,9]. Minimizing risk factors and maximizing the patient's ability to stave off fatal infections require a truly multi-disciplinary effort.

## 2. Risk Factors for Infection

The nature of the burn wound and factors in the patient environment both contribute significantly to the increased risk of infection. Early in the hospital course, patients are at highest risk for Gram-positive infections, usually involving the skin or soft tissue [9]. This results from the thermal injury compromising the body's primary barrier to the external environment [10,11]. Furthermore, unexcised burn eschar is an ideal environment for bacterial growth. The subsequent hypovolemic and hypermetabolic post-burn state coupled with a relative immune suppression and dysregulation leaves patients at risk for developing serious infectious complications [3]. The longer the stay in the hospital, the greater the risk for pneumonia, blood stream infections and urinary tract infections from

Gram-negative or multi-drug-resistant species [7]. Invasive lines and catheters are prime locations for pathogens to enter the body and proliferate. Intubated patients or those with inhalation injury are almost twice as likely to get pneumonia and their risk of dying from pneumonia is increased by one-third [12]. Much of this risk is thought to be caused by decreased mucociliary clearance, airway obstruction and diminished perfusion. However, future research is still needed to uncover many aspects of the pathogenesis of inhalation injury and subsequently its full contribution to infection.

### 3. Pathophysiology of the Burn Wound

The burn wound is defined by three zones. The center of the wound, termed the zone of coagulation, consists of devitalized tissue which has been severely injured from thermal interaction. Next, the zone of stasis is defined by the surrounding tissue just peripheral to the center. This area is ischemic and inflamed with varying depths of burn. Finally, the zone of hyperemia forms the most remote area of the burn wound and contains tissue that is likely to be salvageable [5,6,13,14].

A hyperinflammatory response predominates during the first phase of a burn, with proinflammatory cytokines TNF-$\alpha$ and IL-6 driving the response. Hyperpermeability of the microvasculature is also seen, as histamine stimulates endothelial dysfunction, causing a buildup of fluid in the interstitial space [5,6,13,15].

Along with cytokines and histamine, other mediators released upon thermal injury include reactive oxygen species (ROS) and nitrogen (superoxide anion, hydrogen peroxide, hydroxyl radical, nitric oxide, and peroxynitrite) [5,6,13,14]. Prolonged hypermetabolism can be injurious, if not fatal. ROS release is linked to immunosuppression, inflammatory response syndrome, and multi-organ failure [13]. The previously described events can all be classified under the inflammatory phase of wound healing. The subsequent proliferative phase is marked by an anti-inflammatory response in the body's effort to balance out the initial inflammatory reactions and maintain homeostasis. As anti-inflammatory cytokines IL-4, IL-10 and transforming growth factor (TGF) are recruited, keratinocytes and fibroblasts activate and support revascularization and wound healing [5,13,14]. In the remodeling phase, TGF$\beta$, platelet-derived growth factor (PDGF) fibroblast growth factor (FGF2) and matrix metalloproteinases (MMPs) all aid in wound maturation. The developing scar is formed by collagen and elastin, with the continual conversion of fibroblasts into myofibroblasts [3,5,16].

### 4. Diagnosis of Infection and Sepsis

Recognizing infection and recognizing it early enough to prevent progression to sepsis form a critical facet of burn patient care. Sepsis is defined as "a life-threatening organ dysfunction caused by a dysregulated host response to infection" [17]. Many criteria and definitions have been employed in critical care centers including the systemic inflammatory response syndrome (SIRS) criteria, the sequential organ failure assessment (SOFA) score, the quick sequential organ failure assessment (qSOFA) and the logistic organ dysfunction (LODS) score [18]. All of these have performed with varying degrees of success in non-burn patients [19]. The initial hypermetabolic, inflammatory state of a burn patient causes many of the criteria underlying these early assessment scores (tachycardia, tachypnea, hypotension, fever, and leukocytosis) to be universally present even in the absence of infection. Consequently, these schemes have had limited efficacy for the burn-injured patient [19]. The American Burn Association (ABA), in light of these shortcomings, gathered a panel of experts to come up with a revised set of definitions for infection and sepsis in 2007 (Figure 1), most notably including higher thresholds for SIRS criteria and emphasizing that burn sepsis should be suspected only when there is a change in patient status [20]. Despite these improvements, several studies have subsequently questioned the strength of these criteria. Hogan et al. published a 2012 retrospective analysis of the ABA criteria in 196 burn patients and found that only heart rate and temperature had a significant correlation with bacteremia [21]. Mann-Salinas et al. published a 2013 analysis of the poor efficacy of the

SIRS and ABA criteria in predicting sepsis in burn patients. They identified their own set of predictors (HR > 130, MAP < 60, base deficit < −6, temperature < 36 °C, use of vasoactive medications, and serum glucose > 150 mg/dL) which outperformed existing metrics [22]. Despite significant research and advancement, identifying and defining infection and sepsis in the burn patient population continue to evolve and a universally adopted set of criteria remains elusive. Considering the difficulties of conventional physiologic parameters to recognize infection and early sepsis, many biomarkers have been studied to help aid in early diagnosis of sepsis. Procalcitonin, an endocrine hormone that is undetectable under normal conditions, is released into circulation in the presence of a systemic inflammatory response and bacterial infection. Meta-analysis has shown a sensitivity and specificity as high as 74% and 88%, respectively, in detecting burn sepsis [19]. Furthermore, studies have shown that procalcitonin is superior to other acute-phase proteins (ESR and CRP) in predicting infection and sepsis [23,24].

ABA Sepsis Criteria[a]

| Parameter | Measurement | Notes |
|---|---|---|
| Temperature | ➤ > 39°C or < 36.5°C | |
| Tachycardia | ➤ > 110 beats/minute | |
| Tachypnea | ➤ > 25 breaths/minute *not ventilated | Ventilated > 12L/m minute ventilation |
| Thrombocytopenia | ➤ < 100,000/µL | To start 3 days post-resuscitation |
| Hyperglycemia | ➤ Untreated plasma glucose >200mg/dL OR Insulin resistance <br> • > 7U/h IV gtt <br> • > 25% increase in insulin requirements over 24h | Only in absence of diabetes |
| Inability to continue Enteral feedings >24h | ➤ Abdominal distension, diarrhea, increased residuals | |

* Documented infection by the following criteria
IS required
- Culture positive
- Pathological tissue source
- Clinical response to antibiotics

**Figure 1.** Most recent (2007) consensus guidelines for diagnosis of burn sepsis by the American Burn Association. [a] adapted from Tejiram et al. [20].

Beyond procalcitonin, other frequently studied biomarkers such as CRP, IL-6, IL-8, IL-10, and TNF-$\alpha$ have had much less success predicting infection and sepsis [19,23,25,26]. A recent study by Niggemann et al. showed that pancreatic stone protein (PSP) was the only inflammatory biomarker that reflected a change (3.3–5.5-fold increase within 72 h of sepsis) before the onset of sepsis in severely burned patients [27]. Other studies have shown that abnormally high or persistently elevated levels of biomarkers correlate with poor outcomes and increased probability of infection [19,27,28]. However, given their involvement in many other inflammatory states their predictive utility is still a source of debate.

Further advances in the use of biomarker identification have revolved around using PCR and gene expression (mRNA) profiling to gain a diagnostic advantage over current methodologies. A recent FDA-approved, PCR-based scoring system called Septicyte (Immunexpress, Seattle, WA, USA) uses proprietary biomarker signatures in the early detection of infection as early as 4–6 h after ICU admission [29]. Rather than trying to find a specific pathogen, the test assesses the patient's immune response to suspected infection. As noted previously, the clinical signs of infection and inflammation are often similar, especially in the burn patient. Distinguishing an infection-positive response from an infection-negative inflammatory response can be a significant aid in treatment decisions. Other gene expression-based scoring systems currently being used in critical care settings include the Sepsis Metascore and FAIM3:PLAC8 ratio [30,31]. A 2019 independent analysis comparing the three previously mentioned scoring systems found Sepsis Metascore to be the most accurate in differentiating sepsis from non-infectious SIRS [AUC 0.80 (MetaScore), 0.69 (FAIM3:PLAC8 ratio), and 0.68 (SeptiCyte Lab)] [32].

Lastly, proper antimicrobial treatment depends on the accurate identification of the responsible pathogens. Historically, cultures are obtained and analyzed under the microscope for certain characteristics. This process can be time consuming, and it often takes days for cultures to result. Mass spectrometry, specifically matrix-associated laser desorption–ionization time of flight mass spectrometry (MALDI–TOF MS) is a promising new method for the quick identification of bacteria, yeast and fungi based on their microbial proteins [33]. These microbial proteins create a molecular "fingerprint" or spectra that is stored in a database which can be rapidly referenced. There are currently two databases which have been approved by the Food and Drug Administration (FDA), VITEK MS and MALDI Biotyper. However, many more databases are currently being created [34,35].

## 5. Evolution of the Burn Infection

Burn wound colonization occurs when less than $10^5$ CFU/g are present. Colony counts are performed on cultured wound biopsies as well as histological examination [36]. This low concentration of bacteria usually presents in the absence of a more global, invasive infection, minimal erythema, and no cellulitis [36,37]. When greater than $10^5$ CFU/g are present, it is classified as an infection. Increased pain, poor wound healing, graft loss, erythema and change in odor are subjective signs that many providers rely on to monitor for infection. These subjective signs must be coupled with more objective measurements of systemic infection such as temperature, tachycardia, and increased fluid requirements. However, complicating matters is the pro-inflammatory state of the burn patient in which many of the above clinical signs and symptoms are present even in the absence of infection [38]. Therefore, conventional guidelines for identifying infection and sepsis in non-burn patients do not necessarily perform well for the burn patient. Ultimately, the transition between wound colonization and infection is not well defined. The gold standard for infection detection is wound biopsy and histology; however, cost and limited access can be barriers to using these methods [7].

The burn wound eschar is known to be a nutrient-rich environment easily colonized by microbes that thrive in a protein-rich milieu absent of leukocytes. Wounded, necrotic tissue ceases to carry out the skin's normal barrier functionality and limits the skin's ability to heal by interfering with endothelial cell growth [4,19]. Due to these eschar characteristics, early

excision and grafting have been encouraged to limit the risk for infection and sepsis [39]. Historically, excision and grafting have been performed between 1 and 7 days post-burn, but recent studies have shown benefit from increasingly early intervention. For instance, a recent 2021 study of 836 adult thermal burn patients from the Burns Registry of Australia and New Zealand (BRANZ) hospital sites found that excision within 24 h was associated with reduced length of ICU stay ($6.6 \pm 8.1$ vs. $9.2 \pm 10.6$ days; $p = 0.008$) and lower mean mechanical ventilator hourly use ($94.9 \pm 160.8$ vs. $159.2 \pm 219.1$ h; $p = 0.001$). Other outcome measures including mortality, hospital length of stay and incidence of positive blood cultures were not significantly affected [40]. In instances where early excision is not possible, topical agents such as silver sulfadiazine (Silvadene), mafenide acetate (Sulfamylon) and bismuth subgalactate (Xeroform) help protect the wound from infection and encourage healing [41,42].

Thermal injury initially creates a sterile setting that is colonized approximately 48 h post-injury by either the surviving microbes in the area or by bacteria from healthy skin bordering the eschar. Gram-positive organisms such as *Staphylococcus aureus*, *Corynebacterium*, *Micrococcus*, and *Streptococcus pyogenes* are the first colonizers of the wound bed as these make up the microflora of the skin microbiome [4,19,39]. Further out from injury, infection exposure becomes more endogenous in nature with Gram-negative flora native to the gastrointestinal tract and upper respiratory system taking over the wound bed. Gram-negative bacteria including *Enterobacteriaceae*, *Pseudomonas aeruginosa*, and *Acinetobacter Baumannii* are infectious agents of concern that are translocated from either the digestive tract reservoir or exogenous sources (hands of clinicians, surfaces) [4,39,43]. Infected burn patients treated with various antimicrobial regimens are at risk for developing more serious infections from multi-drug-resistant organisms, yeasts, and even viruses [44,45].

## 6. Agents of Infection

### 6.1. Gram-Positive Bacteria

6.1.1. *Staphylococcus*

*Staphylococcus aureus* is by far the greatest infectious microbe in burn and surgical units across the world. *S. aureus* is a nosocomial microbe that commonly colonizes the human nares and epidermis of approximately 10–35% of the population [46]. *S. aureus* has a variety of virulence factors that make it an effective pathogen. Utilizing quorum sensing, a bacterial method of regulating gene expression through cell–cell signaling, *S. aureus* easily communicates between cells and regulates its virulence based on the environment [47,48]. Adhesins, or cellular appendages, are used by the bacterium for adherence to cells or other biologic surfaces. Once adhered to a cell membrane (using collagen-binding adhesins, elastin-binding protein, fibronectin-binding proteins A and B), *S. aureus* can release β toxin, which degrades sphingomyelins, or α toxin, which is a pore-forming toxin [47,49]. A variety of other toxins including superantigens such as toxic shock syndrome and SE-like proteins can cause serious complications, leading to cytokine storm activation and multi-organ failure [50,51].

With the emergence of methicillin-resistant *S. aureus* (MRSA) and more recently vancomycin-resistant *S. aureus*, there is a need for new antimicrobial treatments. However, current standards of care still recommend the use of drugs such as Daptomycin and Linezolid, which were first discovered decades ago [52]. While these drugs remain effective in most cases, discoveries of new therapeutics have been lacking in recent years. Despite the availability of effective treatment, MRSA remains dominant worldwide, with some burn centers reporting an incidence of greater than 50% [37]. One study showed that among 93 clinical isolates of *S. aureus* isolated over the course of 6 months in Afghanistan, prevalence of methicillin resistance was 66.3% [53].

6.1.2. Streptococcus

Streptococcus bacteria are another toxin-producing bacteria similar to *S. aureus*. *Streptococcus* secrete pyrogenic Group A toxins, which cause a variety of insults such as toxic

shock syndrome, necrotizing fasciitis, scarlet fever, pneumonia, and pharyngitis [50,51,54]. In addition to Group A *Streptococci* serotypes (SPEA, SEC and G to M), streptococcal superantigen, and streptococcal exotoxin ($Z_n$), there is a core chromosome-encoded SPEB which also displays unique virulency [51]. The M protein, endemic to *Streptococcus*, is an antiphagocytic virulence factor which directly influences inflammation through evasion of opsonization [51]. Streptococcus is a bacterium along with *S. aureus* that is implicated in many cases of graft failure in burn patients [37]. In recent decades, the incidence of streptococcus infections has decreased due to the use of aminoglycosides. Additionally, penicillin and other β-lactams have had success treating *Streptococci* [49,50].

### 6.1.3. *Enterococcus*

*Enterococcus* has become an increasingly lethal pathogen (comparable to MRSA in mortality) especially in the immunocompromised burn population [37]. Vancomycin-resistant enterococci (VRE) have been known to transfer their vancomycin resistance to *S. aureus.* Infections with both of these pathogens simultaneously leads to a significant increase in comorbidity [55,56]. Enterococci results in approximately 5% of endocarditis infections in the burn population. It is a significant cause of nosocomial bacteremia which lengthens hospital stays, complicates drug treatment, and significantly increases treatment costs for infected patients [56].

### 6.2. *Gram-Negative Bacteria*
### 6.2.1. *Pseudomonas*

*Pseudomonas* is a major cause of disease and infection, contributing to approximately 2 million infections and 90,000 deaths per year [57]. Between 60% and 90% of blood stream infections (BSIs) occurring in the burn patient are due to Gram-negative organisms. *Pseudomonas aeruginosa* remains the most common Gram-negative agent. BSIs manifesting as sepsis are serious and difficult to treat. *P. aeruginosa* is responsible for 17–21% of endocarditis in burn patients. While quite rare in the general population, endocarditis is much more common in the burn population [58]. *P. aeruginosa* uses several mechanisms to improve its survival but biofilm formation is the most effective. Biofilms are hard to penetrate and eradicate with standard antimicrobial therapy and are able to easily form on any invasive line [49,57]. Burn wounds infected with *P. aeruginosa* are often green/yellow in appearance and have a distinct odor [37]. If these infections are allowed to progress, black/blue lesions can form. Moreover, multi-drug-resistant *P. aeruginosa* prevalence has become a problem for burn clinicians. A recent study showed that in 93 isolates from burn wound infections, 100% of them showed some type of microbial resistance [59].

### 6.2.2. *Acinetobacter*

*Acinetobacter* is an opportunistic bacterium that has been previously described with low virulency. However, *Acinetobacter*'s pathogenicity is especially of concern to immunocompromised patients. For this reason, *Acinetobacter* is clinically relevant in burn as one of the most common nosocomial infections [60]. *Acinetobacter* infection can manifest in the form of pneumonia, wound, skin and soft tissue infections, osteomyelitis, urinary tract infections or endocarditis [60,61]. Additionally, within ICU units, *Acinetobacter* is a common source of HAIs. This coccobacillus has recently developed multi-drug resistance, most notably to Carbapenem antibiotics which have historically been used as a treatment of last resort. Carbapenemases allow *Acinetobacter* to evade antimicrobial treatments with drugs such as imipenem, doripenem, and meropenem. Colistin (Polymyxin E), a drug that was first discovered in 1949 to treat Gram-negative infections and subsequently fell out of favor due to its side effect profile, has been used increasingly in the treatment of multi-drug-resistant *Acinetobacter* [62,63]. While there is some evidence of colistin-resistant strains, they remain relatively rare most likely due to the lack of use of colistin in the last 50 years. Despite its nephrotoxicity and neurotoxicity, colistin remains a viable option of last resort for the treatment of *Acinetobacter*.

### 6.2.3. *Enterobacter*

Several key traits make *Enterobacter* a successful opportunistic bacterium and cause of a variety of nosocomial infections in the hospital setting. *Enterobacter* spp. are equipped with flagellum for motility, can form biofilms, and possess endotoxins, exotoxins, alpha hemolysins, and cytotoxins. *Enterobacter* is implicated in wound and urinary tract infections, pneumonia, meningitis, and septicemia. Most clinical manifestations are in immunocompromised individuals including burn, diabetes, cancer, and premature infants [64].

### 6.3. Fungi and Yeasts

The most commonly isolated fungal species are *Candida albicans* and *Aspergillus* species. With the introduction of broad-spectrum antibiotics in the 1960s, bacterial incidence and prevalence decreased, while fungal infection incidence increased by a factor of 10 [65]. Though fungal infections are more common in burn patients with higher TBSA (>40%), recently it has been reported that infection, especially with *Candida* species, is not specific to TBSA, age, or inhalation injury [37]. On the other hand, infections with molds other than *Candida* (which exists as part of the innate microbiome) are more significant causes of morbidity and mortality in burn [4,37,65]. Zygomycetes, a fungus present in soil, can cause mortality rates of up to 54% [65]. Treatment of fungal infection involves rapid excision of infected tissues, autografting, and the use of topical anti-fungals such as clotrimazole and nystatin [4,37]. Systemic approaches to curbing fungal infection include fluconazole and amphotericin B. An emerging treatment for fungal infections, chiefly for drug-resistant *Candida albicans*, includes echinocandins such as caspofungin. For non-*Candida* species including *Aspergillus*, Zygomycetes, and *Fusarium*, voriconazole is used [37,65].

It is worth mentioning that current diagnostic methods for fungal infections have poor sensitivity and a high rate of false negatives, leading to delays in diagnosis. Biomarkers of fungal infection such as serum $(1 \rightarrow 3)$-B-D-glucan (BG) have been touted as successful markers of invasive fungal infection. Specifically, the Fungitell assay (Pyrosate, Associates of Cape Cod, Inc., Falmouth, MA, USA) has been used to measure BG levels in critically ill patients [66,67]. However, many different factors can influence BG levels (bacteremia, use of gauze bandaging, blood transfusions), leading to false positives. A 2011 study in 21 burn patients found elevated BG levels at baseline for 50% of patients, none of which went on to develop invasive fungal infection [68]. Therefore, the Fungitell assay in burn patients should be used with caution.

### 6.4. Viruses

Innate and adaptive immune dysregulation is the cause for viral infection in burn populations. Viral infections present as primary, latent, or nosocomial. Primary infection results when a viral pathogen invades a host that has no prior immunity built up to protect against such a pathogen. Latent viruses can be reactivated in the host after the first week of injury due to immune dysfunction. Nosocomial infection from an agent the host has already established immunity to is not uncommon in ICU settings. Relevant viruses include varicella zoster virus, herpes viruses, cytomegalovirus, hepatitis, HIV and SARS-CoV-2 [69].

## 7. Specific Infections in the Burn Patient

While wound infections are most common, given the chronically immunosuppressed state of burn patients, they are also at increased risk for many systemic infections including catheter-associated urinary tract infections (CAUTIs), central line-associated blood stream infections (CLABSIs), and pneumonia. The risk for these invasive infections is positively correlated with the size (TBSA > 30%) and depth of the burn [44]. Larger burn injuries often require longer hospital stays as well as more invasive procedures. Hospital-acquired infections (HAI) can develop when the external exposome of the hospital environment becomes a source of transmission. Furthermore, it can be difficult and time intensive to track HAIs and insufficient monitoring contributes to the multitude of infectious factors [44].

### 7.1. Pneumonia

Pneumonia is the leading systemic infection experienced in the burn patient population, complicating up to 2.2% of admissions [8]. While pneumonia is a risk and a challenge for any critical care patient, burn patients are more vulnerable due to their relative immunocompromised state. The presence of inhalation injury with or without large TBSA burns often necessitates the need for patients to be on a ventilator, further increasing the risk of ventilator-associated pneumonia. Recent studies have shown that the risk of VAP increases with burn size, presence of inhalation injury, prolonged mechanical ventilation, and longer ICU stays. VAP has also been shown to increase mortality [19]. Specific prevention bundles that include elevating the head, oral care, ulcer and DVT prophylaxis, subglottic suctioning, daily spontaneous breathing trials and daily sedation interruption have shown success in limiting the incidence of VAP by 44% to 54% [70]. Fiberoptic bronchoscopy has also emerged as the standard of care in the diagnosis and treatment of VAP. In patients with inhalation injury complicated by pneumonia, studies have shown a reduction in length of ICU stays, ventilator days and hospital stays in patients who have undergone bronchoscopy compared to those that did not [71].

### 7.2. Catheter-Associated Urinary Tract Infections (CAUTIs)

Catheter-associated urinary tract infections are the third most common complication in burn patient hospitalizations and complicate up to 2.4% of burn patient hospital stays [8]. The national health care safety network reported rates of CAUTIs almost 7-fold higher in burn critical care units (1.2 infections/1000 catheter days vs. 7.4 infections/1000 catheter days) [72]. This is most likely due to a multitude of factors including burn patient susceptibility to infection, decreased burn patient mobility and the need for both large and accurate titration of resuscitation volumes requiring frequent and extended use of Foley catheters. The longer a Foley catheter is in place, the higher the chance of infection. Current guidelines recommend the diagnosis of UTI if the patient has a fever greater than 39.5 °C, urgency, frequency, dysuria or suprapubic tenderness and a urine culture with greater than $10^5$ CFU/mL with no more than two species of organisms [17]. Most importantly, the presence of pyuria alone is not an indication for antibiotic treatment and the absence of pyuria in a symptomatic patient suggests an alternative diagnosis [73]. Prevention bundles emphasizing proper aseptic technique during catheter insertion and maintenance and prompt removal of Foley catheters have been shown to significantly reduce the incidence of CAUTIs in burn patients [74].

### 7.3. Central Line-Associated Blood Stream Infections (CLABSIs)

Central line-associated infections remain an issue for burn patients and critical care patients in general despite multiple studies examining placement techniques, use of topical antibiotics, frequency of exchanges and location of insertion. There is a wide range of institutional practices and guidelines and much of the research disagrees on an optimal approach. For instance, one study of 32 patients with an average TBSA of 58% found that topical mupirocin, in conjunction with other safe insertion practices, significantly reduced the incidence of CLABSIs (5.3 vs. 29.1 per 1000 catheter days, $p < 0.001$). Furthermore, the study did not find any difference in risk based on the anatomical placement of the central line (internal jugular vs. femoral vs. subclavian) [75]. This contrasts with multiple studies that have shown an infection difference in line placement locations [76–79]. Multiple studies have agreed that CLABSIs are positively correlated with larger %TBSA and that placement of the catheter through burned skin increases the risk of infection by as much as a factor of 4 [77,79].

## 8. Treatment and Prevention

### 8.1. Pharmacokinetics and Pharmacodynamics of Antibiotics

Antibiotic selection and dosing strategies are of particular interest in burn patients because of the unique environment created by their systemic inflammatory state. Car-

diovascular, renal, respiratory, and hematological dysfunction are common and lead to frequent changes in peripheral perfusion, endovascular permeability, and fluid balance [80]. Conventional treatment regimens used in non-critically ill patients are unlikely to be effective or sufficient. Furthermore, antibiotic resistance is a growing problem for burn patients who often become infected with multi-drug-resistant organisms (MDROs) [81]. Identifying optimal antibiotic treatment algorithms can help mitigate this threat and aid in the development of new antibacterial drugs.

Early antibiotic treatment of infection has shown to be essential for positive outcomes in the burn patient population [82,83]. However, current standard of care guidelines discourage the use of antibiotics on admission due to the rise of MDROs. Despite some surgical guidelines advocating for the use of pre-operative antibiotics, recent studies suggest that withholding antibiotics does not place the patient at any greater risk [84]. It should be emphasized that the timing and dosing of antibiotics need to be tailored to each patient and cannot be calculated without an in-depth understanding of the pharmacodynamic (PD) and pharmacokinetic (PK) properties of the drugs.

Pharmacokinetics (PK) is the change in concentration of a drug in the body over time. In other words, what the body does to the drug. Pharmacodynamics (PD) focuses on the physiologic effect of the drug, or what the drug does to the body. A primary endpoint of PD is the minimum inhibitory concentration (MIC), that is, the minimum concentration of a drug which prevents bacterial growth in vitro. PK centers around the distribution, absorption, metabolism and elimination of a drug [85]. Distribution describes the process by which a drug is distributed throughout the various tissues and organs of the body. Absorption describes how much of the drug is delivered into the systemic circulation. Metabolism describes the conversion of the drug between active and inactive forms. Elimination focuses on the removal of the drug from the body either via the renal system or through certain non-renal pathways such as hepatic metabolism or direct loss of substrate through open wounds. There is a complex PK/PD interplay that describes the optimal drug exposure required for maximal bacterial killing [86].

Due to the dynamic and everchanging nature of a burn patient's hospital course, clinicians must be cognizant of the precise physiologic state of their patients and how this might influence drug selection and delivery. Pharmacokinetic properties, for instance, are increasingly important during the initial resuscitation phase. Large volumes of fluid cause changes in the distribution of the drug and subsequently require larger doses to have the desired effect. However, concurrent changes in elimination via the kidneys and altered hepatic metabolism can cause undesired side effects from too much accumulation of a drug. Studies have advocated for more frequent or continuous dosing of antibiotics to optimize PK parameters. A randomized controlled trial examining the effects of continuous vs. intermittent dosing of beta-lactam antibiotics in critically ill patients found significant increases in clinical cure rates in the continuous groups [87]. This finding was most likely due to the fact that continuous dosing is able to achieve higher plasma concentrations of the drug. Another study found that real-time, therapeutic drug monitoring of plasma concentrations, despite increased cost and complexity, led to more precise dosing but did not affect outcomes [88]. Future research regarding TDM and outcomes in burn patients is still ongoing.

### 8.2. Phage Therapy

While antibiotics are the gold standard in the treatment of infection, emerging bacterial resistance and a limited pipeline of new drugs have led to the search for novel treatments. Phage therapy, the practice of using naturally occurring bacterial viruses (phages) to treat bacterial infection, has been around for decades [89,90]. However, phage therapy fell out of favor given the relative complexity and lack of knowledge compared to antibiotics. Recent positive research in animal models has advocated for phages as an alternative or supplement to help bolster the waning effectiveness of antibiotics [91,92]. Despite its promise, the re-emergence of phase therapy is still in its relative infancy. Therefore,

implementation is hindered by a lack of information regarding their efficacy or guidelines for their use or formulation [93–95]. Nevertheless, it could be an important adjunct for treating infections in the future.

*8.3. Minimizing Contamination*

Understanding the intricacies of antibiotic usage is useful not only for infection control and treatment but for preventing contamination of the patient as well. Disinfection and isolation strategies for burn patients are wide-ranging, with most institutions designing their own protocols for infection control. Recent decades have seen the rise of dedicated burn units, separate from the rest of the population, specialized burn operating rooms, increased personal protective equipment for providers and patients, and better wound care [20]. Despite these advances, hospitals and burn units remain a nidus for pathogens that can severely affect burn patient outcomes.

Infection control measures must consider the risk of infection from both endogenous and exogenous sources. Exogenous sources come from the environment, or anything outside the body. Endogenous sources come from the patient, usually in the form of respiratory secretions or open wounds. Endogenous sources are much more common sources of infectious outbreaks than exogenous ones [96]. However, exogenous sources still account for 20–25% of hospital-acquired infections and can arise from countless locations, making it particularly important to maintain strict disinfection protocols [97]. A few of the more common sources are medical equipment (ultrasound probes) and monitoring devices ($O_2$ probes, ECG leads). Not surprisingly, sinks, toilets and wash tubs are also high on the list [20].

New-age decontamination devices are now starting to become more commonplace. Hydrogen peroxide is a powerful decontaminate that is now being used between patient stays to disinfect rooms. Hydrogen peroxide is employed either in aerosolized or vapor form and can disinfect a room in a matter of hours. These systems have had success against MDROs such as MRSA and VRE [98]. Another promising solution involves the use of cold plasma. Conventional antimicrobial agents such as povidine iodine and chlorhexidine are excellent at killing bacteria but have cytotoxic effects on healthy cells as well. Cold plasma, on the other hand, has the ability to kill microbes without harming healthy tissue. Cold plasma is created by adding heat to an inert gas. The resulting partially ionized gas is made up of many ions, electrons, ultraviolet photons and reactive oxygen species that exhibit antimicrobial effects. It can be applied to the wound bed too for disinfection and also stimulates cell migration and proliferation to promote wound healing [99–101].

Ultraviolet light (UV) systems have also been employed in an effort to disinfect patient rooms. Multiple studies have shown that a particular type of UV light (UV-C) is germicidal and can successfully decontaminate a room to the point where future patients are subsequently less likely to acquire any infective pathogens. The only downside to these systems is that they cannot be employed while patients are in the room [98].

Visible, blue light has also been studied as a bactericidal method of decontamination and has shown success in reducing the amount of surface bacteria in a room. While there are limitations to this method against certain pathogens (*C. difficile*), this type of light can be employed even during patient encounters [102].

Surfaces in high-traffic areas such as health care facilities are a serious source of spread of infection and HAI have been associated with poor long-term outcomes [103]. Recent studies have shown similar spread of infection from health care workers touching a patient's skin and touching surfaces in their rooms [104]. Self-disinfecting surfaces coated with heavy metals (copper and silver) or impregnated with germicide (Triclosan and Goldshield) are now being studied as a way to minimize this spread and have shown success against problematic pathogens such as *C. difficile*, MRSA and *E. coli* [105].

## 9. Conclusions

Understanding every facet of infection in the burn patient is imperative to providing them with a high level of care due to their increased susceptibility, their relative immuno-compromised state and their potential for poor outcomes. Innovations in the diagnosis, treatment and prevention of infection in recent decades have continued to improve morbidity and mortality for burn patients. However, infection remains the most common cause of mortality in the burn patient and will only continue to evolve over time.

**Author Contributions:** Conceptualization, E.J.K., J.W.S. and M.A.O.; methodology, E.J.K. and M.A.O.; writing—original draft preparation, E.J.K. and M.A.O.; writing—review and editing, E.J.K., M.A.O., B.C.C. and J.W.S. All authors have read and agreed to the published version of the manuscript.

**Funding:** This research received no external funding.

**Institutional Review Board Statement:** Not applicable.

**Informed Consent Statement:** Not applicable.

**Data Availability Statement:** Not applicable.

**Conflicts of Interest:** The authors declare no conflict of interest.

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
