# Peer review of "Infection and Burn Injury"

_2673-1991, doi:10.3390/ebj3010014_

Round 1

Reviewer 1 Report

Overall, this is an excellent topic choice for review that is of great interest to the field, and will be somewhat useful as other recent reviews on this subject are behind paywalls and this one will be open access. The manuscript is well written and encompasses a wide spectrum of the topic. Unfortunately the scope of the manuscript is so wide that the discussion of each important topic is so superficial it is barely useful. I wish this work was several papers each one addressing a section discussing the literature in depth. 

Author Response

Thank you for your review and interest in this work.  As you point out, there have recently been reviews on similar topics.  An entire issue of Surgical Infections was devoted to “Infections in Burn Patients” (Vol. 22, Issue 1) in February of 2021. Reproducing a work such as that is not the present ask or goal. However, we agree that in trying to include all facets of such a broad topic it might have caused us to be too brief in our handling of some of the material. In the presently revised manuscript, we have tried to add additional content to several sections (i.e. diagnosis, fungal infections, treatments) in an effort to make certain areas more complete and thorough.

Reviewer 2 Report

Infections and Burn Injury is an ambitious, general review of a vast topic. It fits very well to the focus of European Burn Injury and is informative and mostly well understandable to even a non-burn professional. I have some remarks I hope the authors would address:

  • Line 77 or actually Fig.1: Tachypnea over >110 and not <110.
  • Line 130: I do not think silversulphadiazine is any more a first line topical agent in burn wound treatment. Also, early excision, to me, is nowadays definitely earlier than 7 days, see:

Moussa A, Lo CH, Cleland H. Burn wound excision within 24 h: A 9-year review. Burns. 2021 Sep;47(6):1300-1307.

  • Line 152: quorum sensorum is not familiar for many of the readers. Please make a short description of what it means.
  • Line 162:s: To my knowledge, true new antibiotics against gram positive germs has not been developed in last decenniums and there is nothing really new coming neither.
  • Line 214: What about colistin and Acinetobacter?
  • Line 379: Cold plasma. This is very new approach and hardly anyone is aware of this option. Most people connect it with blood plasma (like I did!) which means it should be explained clearly.

When discussing future possibilities to treat or prevent infections phage therapy should be mentioned. It seems to be coming and is, already here, again. Also, vaccination against some gr positive bacteria etc. may be one option. Better, faster and much more accurate diagnostics will be one very important tool, e.g. MALDI-TOF and RNA sequencing.

Author Response

Thank you for your review and comments.  We have incorporated the changes you suggest as indicated below, and hope you find that they have improved the manuscript.

1) Tachycardia corrected to >110 in table

2) included reference to early excision in the first 24 hours and modified language to show early excision and grafting now being done sooner. Also added a couple topical agents other than silver sulfadiazine that can be used.

3) included a definition of quorum sensing

4) included comment on lack of new antibiotics in last few decades

5) added a few sentences about colistin and treatment of MDR acinetobacter

6) added a better description of cold plasma

7) added a paragraph regarding phage therapy

8) added a paragraph on MALDI-TOF

Round 2

Reviewer 1 Report

Thank you for improving the paper. I am still confused based on the title about whether this is a paper about burn wound infections or infection and burn injury as the title states. If the latter, a more global presentation of the subject in the introduction would be useful. For instance, the first sentence, ...burn care in the past few decades has allowed significant decreases in mortality.. is overused, probably represents a period of almost 40-50 years now, and probably should be retired. It would be more helpful to know what contribution infection has to overall mortality in burns.

Before going into diagnosis, what are the risk factors for infection after burns? How does one identify someone that is at high risk for infection? How has the old paradigm of acute systemic inflammatory response followed up by a second hit changed? What is the current thought on how some patients breeze through their burn injury and others get beaten down by repeated infections and obvious prolonged immunosuppression? 

How does inhalation injury contribute to infection?

What are the risk factors for wound infection? Any clinical pearls? Such as does the post burn day that the cellulitis appears suggest a particular organism (i.e. strep vs staph)?

Author Response

Thank you for reviewing the revised manuscript. I've attempted to address all your questions.

1) I have re-worked the opening paragraph and taken out that specific sentence to try to better explain the scope of the paper. Our aim was for this paper to be about burn infections in general and not just wound infections.

2) I have added details to address these points if not already addressed

3) I added some detail regarding inhalation injury and noted that it is still an area for future research

4) I have elaborated on risk factors and timing of certain infections.